# PET Oncological Radiopharmaceuticals: Current Status and Perspectives

**DOI:** 10.3390/molecules27206790

**Published:** 2022-10-11

**Authors:** Mai Lin, Ryan P. Coll, Allison S. Cohen, Dimitra K. Georgiou, Henry Charles Manning

**Affiliations:** 1Cyclotron Radiochemistry Facility, The University of Texas MD Anderson Cancer Center, Houston, TX 77054, USA; 2Department of Cancer Systems Imaging, The University of Texas MD Anderson Cancer Center, Houston, TX 77030, USA

**Keywords:** PET, radiopharmaceuticals, theragnostics

## Abstract

Molecular imaging is the visual representation of biological processes that take place at the cellular or molecular level in living organisms. To date, molecular imaging plays an important role in the transition from conventional medical practice to precision medicine. Among all imaging modalities, positron emission tomography (PET) has great advantages in sensitivity and the ability to obtain absolute imaging quantification after corrections for photon attenuation and scattering. Due to the ability to label a host of unique molecules of biological interest, including endogenous, naturally occurring substrates and drug-like compounds, the role of PET has been well established in the field of molecular imaging. In this article, we provide an overview of the recent advances in the development of PET radiopharmaceuticals and their clinical applications in oncology.

## 1. Introduction

Molecular imaging is the visual representation of biological processes that take place at the cellular or molecular level in living organisms. The main advantage of molecular imaging is that information is acquired for the characterization and quantification of biological processes with negligible perturbation to the organism [1,2]. While the term molecular imaging was proposed in 2001 [1,2], the concept behind molecular imaging was first introduced in the early 1900s [3] with the use of ^131^I for the imaging of recurrent thyroid carcinoma. Today, molecular imaging is regarded as a multidisciplinary practice [1,2], and a driver in the transition from conventional medical practice to precision medicine. Sometimes known as “personalized medicine”, precision medicine is an innovative approach to tailoring disease prevention and treatment with the goal of targeting the right treatments for the right patients at the right time [4]. The idea of precision medicine has been growing in clinical research for the past decade and molecular imaging has played a vital role in the advancement of this field.

One of the most popular molecular imaging techniques is positron emission tomography (PET). To perform a PET imaging study, a radiotracer bearing a positron-emitter is injected into a living organism. As the radioisotope undergoes positron decay, the emitted positron travels in the surrounding tissues. When the positron loses its kinetic energy, it interacts with an electron and undergoes annihilation producing a pair of 511 keV photons moving in approximately opposite directions which are detected by the PET instrument. Thus, the technique depends on simultaneous or coincident detection of these photons. Modern PET scanners have the ability to obtain absolute quantitative imaging data (e.g., Bq/mL) after corrections for photon attenuation and scattering. However, since most clinical PET scanners have limited spatial resolution, 6–8 mm for humans and 1 mm for small animals [5], PET needs the addition of anatomical CT or MRI to localize the functional information. PET has the unique ability to image naturally occurring compounds via labeling with ^11^C, ^13^N, or ^15^O to produce compounds that are chemically and biologically identical to those found in living organisms. PET can also be utilized to image other molecules through the synthesis of analogues bearing radioisotopes.

In this article, we provide an overview of the recent advances in the development of PET radiopharmaceuticals and their clinical applications in oncology. All the radiopharmaceuticals presented herein have been approved by the FDA for routine clinical practice and/or entered clinical trials. The references are selected according to the needs of the introduction of the tracer and the breakthroughs that facilitated the tracer from bench to bedside. As PET imaging is intricately linked to many disciplines such as biology, radiochemistry, and nuclear physics, a thorough understanding of many principles in different areas is often necessary for the development of radiolabeled compounds. Consequently, a brief discussion of certain aspects of preclinical radiopharmaceutical development is also included to provide a contextual understanding of this emerging field. While the present discussion is limited primarily to cancer cell-targeted approaches, it should be emphasized that PET and radiopharmaceuticals are critical tools enabling breakthroughs and clinical care across a variety of other disease settings.

## 2. Current PET Radiopharmaceuticals beyond [^18^F]FDG

The design of a PET radiopharmaceutical depends on the pharmacological pathway being imaged. Therefore, certain radiopharmaceuticals are designed to target receptor proteins or metabolic pathways, e.g., [^68^Ga]Ga-PSMA-11 for targeting prostate-specific membrane antigen and [^11^C]methionine for imaging amino acid metabolism, while others are designed to target physical processes, e.g., [^15^O]H_2_O and [^82^Rb]RbCl_2_ for the evaluation of changes in blood flow. An ideal PET radiopharmaceutical must demonstrate high specificity, which can be achieved by rapid localization in target tissues and fast clearance from non-target organs (e.g., liver and kidney). Because PET is dependent on the detection of a radioisotope, the selection of radionuclide should balance the need for in vivo stability during the imaging with considerations of radiation exposure. Ideally, the physical half-life of a radionuclide and the biological half-life of the molecule to be labeled should be compatible. If the half-life of the radionuclide is too short, it would be challenging to complete the biological evaluations of the radiopharmaceutical. On the other hand, a long half-life will result in unnecessary radiation exposure.

Among all radiopharmaceuticals for PET imaging, the glucose analogue [^18^F]fluorodeoxyglucose ([^18^F]FDG) is the most widely used to date in oncological applications. The molecular targets of [^18^F]FDG are glucose transporters and hexokinase [6,7]. [^18^F]FDG follows a metabolic pathway similar to glucose in vivo. However, unlike glucose, which is metabolized to carbon dioxide and water, [^18^F]FDG cannot fully proceed through glycolysis and is instead trapped within cells. While [^18^F]FDG shows an increased uptake in cancer relative to normal tissue due to the increased metabolism of cancer cells, [^18^F]FDG is unable to target cancer cells specifically as all cells utilize glucose. Physiological conditions such as inflammation and infection can also cause high [^18^F]FDG uptake in affected tissues and lead to false-positive diagnoses [8,9]. Consequently, developing other radiopharmaceuticals with better targeting capability for cancer imaging is warranted.

### 2.1. Nucleotide Analogues for Imaging Rapid Cell Proliferation and T-Cell Activation

Rapid cell proliferation is a classic characteristic of cancer cells that leads to an elevated rate of DNA replication. Imaging of cell proliferation has long been a goal of nuclear imaging research, with most of the effort focused on radiotracers that can be incorporated into DNA synthesis. Ex vivo assessments of cell proliferation in human tissues were first reported in 1960 with [^3^H-methyl]thymidine ([^3^H-methyl]TdR) [10].

Along with the increased use of PET technology, [^11^C-methyl]thymidine ([^11^C-methyl]TdR, Figure 1) was developed to image tumor proliferation [11]. Due to the short half-life of ^11^C (T_1/2_: 20 min) and the rapid catabolism of thymidine, analogues that are resistant to in vivo degradation have been developed and labeled with isotopes possessing longer half-lives. Following the development of [^11^C-methyl]TdR, 3′-deoxy-3′-[^18^F]fluorothymidine ([^18^F]FLT, Figure 1) and 1-(2′-deoxy-2′-[^18^F]fluoro-1-β-D-arabinofuranosyl)-thymine ([^18^F]FMAU, Figure 1) have been successfully used to image cell proliferation. Although [^18^F]FLT is not incorporated into DNA after phosphorylation, its uptake is correlated with tumor cell proliferation due to the phosphorylation and trapping by thymidine kinase 1 which reflects the rate of DNA replication [12]. Selective uptake of [^18^F]FLT in tumor cells over inflammatory cells suggests that this tracer will be superior to [^18^F]FDG for the selective detection of malignant cells [13]. Our group has also demonstrated that [^18^F]FLT PET closely reflects pro-survival responses to targeted therapy that are mediated by PI3K-mTOR activity [14,15,16]. Because the activation of pro-survival mechanisms forms the basis of numerous modes of resistance, [^18^F]FLT PET may serve a critical role in predicting tumors that exhibit molecular features that tend to reflect recalcitrance to mitogen-activated protein kinase (MAPK)-targeted therapy.

In addition to thymidine-based radiopharmaceuticals for imaging cell proliferation, a guanosine analogue, 2′-deoxy-2′-[^18^F]fluoro-9-β-D-arabinofuranosylguanine ([^18^F]FAraG, Figure 1), has recently shown potential to monitor activated T-cells in a preclinical study following oxaliplatin/cyclophosphamide treatment of mice bearing xenograft tumors [17], as guanine ribonucleotide plays an important role in T-cell activation [18,19]. Clinical trials with this radiotracer are currently underway to verify this finding in patients undergoing treatment with immunotherapy and/or radiation therapy [20].

### 2.2. Amino Acid Analogues for Imaging Protein Metabolism

Protein synthesis is a fundamental process for all cellular functions. Due to an accelerated growth rate in cancer cells, the demand for amino acids (the building blocks of proteins) is elevated as well. Therefore, it is expected that imaging of radiolabeled amino acid analogues could provide information regarding protein metabolism of malignant cells.

Among all radiolabeled amino acids, [^11^C-methyl]-l-methionine ([^11^C]MET, Figure 2) is currently the most well-established and utilized amino acid radiopharmaceutical. [^11^C]MET PET has been shown to be useful in delineating ependymomas, medulloblastoma, and astrocytomas in pediatric patients and can effectively differentiate between radiation-induced brain tissue injury and tumor recurrence [21,22,23,24]. However, the short half-life of ^11^C limits the clinical implementation of [^11^C]MET PET to highly specialized imaging centers with onsite cyclotron facilities. Consequently, ^18^F-labeled amino acid radiopharmaceuticals (T_1/2_: 110 min) such as [^18^F]FET, [^18^F]FDOPA, and [^18^F]FACBC, (Figure 2) have been developed to overcome this logistical limitation. Currently, both [^18^F]FDOPA and [^18^F]FACBC (Axumin^®^) have been approved by the FDA to aid disease monitoring for people with suspected parkinsonian syndromes and prostate cancer recurrence, respectively [25,26]. On the other hand, while [^18^F]FET is still under clinical trials, the FDA has granted Orphan Drug Designation (ODD) for [^18^F]FET to image patients with glioma [27].

Recently, a glutamate analogue tracer, (*S*)-4-(3-[^18^F]fluoropropyl)-L-glutamic acid ([^18^F]FSPG, Figure 2), is starting to receive great attention as it has been shown to specifically target the cystine/glutamate antiporter (xc^−^)—a biomarker that is frequently overexpressed in cancer and several neurological disorders [28]. Pilot studies examining the dosimetry and biodistribution of [^18^F]FSPG in healthy volunteers [29,30] and tumor detection in patients with non-small cell lung cancer (NSCLC), hepatocellular carcinoma, and brain tumors showed promising results [31,32,33,34,35,36,37]. The low uptake in the brain, lung, and bowel makes [^18^F]FSPG an excellent imaging agent characterized by high tumor-to-background ratios (Figure 3) [38]. The development of reliable and cGMP-compliant procedures for [^18^F]FSPG production will help increase the clinical adoption of this radiotracer. Our facility has newly developed automated approaches using commercially available synthesizers to mitigate this issue and warrant the large-scale production of [^18^F]FSPG [39]. This fully automated process is able to produce [^18^F]FSPG at the Curie level and requires less than 40 min with radiochemical yields ranging from 13–35% (non-decay corrected) and radiochemical purities of >90%.

Continuing the success of the amino acid radiopharmaceuticals that are previously mentioned, we and others have also demonstrated that both ^18^F- and ^11^C-labeled glutamine analogues (Figure 2) hold great potential in clinical applications [40,41,42,43] as glutamine represents an important metabolic substrate that is dysregulated in cancer [44]. Our first-in-human PET imaging of [^11^C]glutamine in patients with metastatic colorectal cancer indicated that [^11^C]glutamine PET appears safe for human use and allows noninvasive visualization of metastatic colon cancer lesions in multiple organs (Figure 4) [40]. In addition, other research groups have also demonstrated the potential of (2*S*,4*R*)4-[^18^F]fluoroglutamine ([^18^F]4F-Gln) in tracking cellular glutamine pool size in breast cancers with differential glutaminase (GLS) activity and detecting increases in cellular glutamine pool size induced by GLS inhibitors [45,46]. Further studies are underway to elucidate the potential of these two tracers for other cancers and for monitoring the response to treatment.

### 2.3. [^11^C]Choline and [^18^F]Fluorocholine for Imaging Cell Membrane Metabolism

Choline is used for multiple essential functions in cells, including being a key component of phospholipids in cell membranes. Thus, choline is present in both benign and malignant processes. Tissues with increased metabolism will have an increased uptake of choline. The uptake of choline appears to be driven by the activity of choline kinase [47]. As a result, cancer cells show an upregulation of choline kinase enzyme activity due to their increased demand for building blocks needed for cell membrane synthesis. High levels of choline and its metabolites in cancer cells have been demonstrated in magnetic resonance spectroscopy (MRS) studies [47]. [^11^C]choline and [^18^F]fluorocholine are useful for the detection and differential diagnosis of various cancers [48,49,50,51]. In fact, PET imaging with [^11^C]choline and [^18^F]fluorocholine was reported to be superior to [^18^F]FDG for patients with multiple myeloma and bone metastasis from prostate cancer [51,52,53]. These tracers provided additional detection of up to 75% more lesions to patients with multiple myeloma, thus, providing evidence of their greater sensitivity in some cancers [51,52].

### 2.4. Nitroimidazole and ATSM-Based Radiopharmaceuticals for Imaging Hypoxia

Hypoxia is a distinguishing feature present in solid tumors and is the result of low oxygen availability in the tumor microenvironment. Not only does the lack of oxygen itself result in resistance to radiation therapy by preventing the formation of reactive oxygen species that can lead to permanent DNA damage, but hypoxic conditions also lead to the expression of hypoxia inducible factors (HIFs), resulting in further mechanisms of resistance [54]. As the early detection of hypoxic regions in tumors is crucial for developing a more personalized approach to treatment, tracers for the imaging and spatiotemporal analysis of these distinctive regions by PET are in high demand [55]. Four commonly used PET tracers for clinical determination of hypoxic regions are [^18^F]FMISO, [^18^F]FAZA, [^18^F]HX4, and [^60/61/62/64^Cu]Cu-ATSM) (Figure 5).

The development and study of [^18^F]FMISO was inspired by the observation of misonidazole molecules’ ability to internalize in cells via passive diffusion due to their lipophilic nature. Though this diffusion can also occur in normal cells, under hypoxic conditions, the nitro substituent on the imidazole ring is reduced by nitroreductases. The resulting hydryoxylamino product has traditionally been thought to covalently bind to macromolecules in the cell, resulting in the eventual accumulation of misonidazole in hypoxic regions [56]. Since its first clinical evaluation in 1992, [^18^F]FMISO has been applied to imaging hypoxic volumes in multiple cancers [57]. Much success has been reported in using [^18^F]FMISO as a prognostic indicator for patients at a high risk of tumor recurrence during radiochemotherapy for head and neck cancers (HNC) [58,59,60] as well as for the mapping of hypoxia in gliomas [61] (Figure 6). However, despite the successes of [^18^F]FMISO PET in monitoring patients with head and neck, breast, or brain cancers, the application of [^18^F]FMISO in other cancer types is limited due to its relatively low tumor-to-background ratios [62,63,64]. This is due to [^18^F]FMISO’s relatively high lipophilicity, which results in a slower clearance rate from normoxic tissue. Such findings inspired the subsequent development of hypoxia-targeting PET tracers with more favorable biokinetic profiles.

Among the nitroimidazole-based ^18^F PET tracers to follow [^18^F]FMISO, [^18^F]FAZA and [^18^F]HX4 have been subject to extensive clinical studies. The synthesis of [^18^F]FAZA was first described in 1999 [65]. Initial PET imaging studies of patients with HNC using this tracer were published in 2007 in order to guide radiation treatment planning [66,67]. The prognostic value of [^18^F]FAZA uptake in tumors of the head and neck was supported in a trial of 40 patients in which the detection of hypoxic regions was associated with a poor outcome [68]. Along with further studies in hypoxia detection using [^18^F]FAZA PET/CT in HNC [69,70], hypoxia has been successfully detected using this tracer in NSCLC [71,72,73], gliomas [74,75], and pancreatic cancer [76]. Due to its faster clearance rate compared to [^18^F]FMISO, PET imaging with [^18^F]FAZA offers advantages such as the ability to image sooner post-injection with less background radiation detected in blood. However, as it is renally cleared, background activity can still be observed in the bladder 4 h post-injection [77]. This drawback was observed in a clinical study testing the feasibility of [^18^F]-FAZA PET/CT to visualize hypoxia in rectal cancer [78]. Failure to observe a significant change in signal between tumors and healthy tissue has also been reported for prostate cancer [79] and renal cell carcinoma [80]. With similar advantages present in [^18^F]FAZA, [^18^F]HX4 is known for its increased water solubility and fast clearance from normoxic tissue. A biodistribution study published in 2010 demonstrated that 45% of injected activity was eliminated from the bladder in 3.6 h [81]. A phase 1 trial was reported in the same year in which the uptake of [^18^F]HX4 was confirmed in patients with lung, thymus, and colon cancer [82]. While comparing image acquisition parameters and results between [^18^F]HX4 and [^18^F]FMISO, Chen and coworkers found that [^18^F]HX4 showed greater specificity towards hypoxic volumes. Unlike [^18^F]FMISO, [^18^F]HX4 can be used for imaging GI tract tumor and allow data collection to start at an earlier time point [83]. Most clinical experiments with [^18^F]HX4 have been performed in patients with either HNC [84,85,86,87] or NSCLC [88,89,90]. These investigations emphasized the value in using tracers such as [^18^F]HX4 to monitor changes in hypoxic volumes during early treatment for making necessary therapeutic adjustments (Figure 7).

PET tracers featuring radioactive copper have also been developed, with [^60/61/62/64^Cu]Cu-ATSM exhibiting much success. This tracer has been suggested as a promising alternative for imaging hypoxia due to its high membrane permeability and low redox potential [91]. In 2000, the first human study featuring a positron-emitting [^62^Cu]Cu-ATSM complex was published [92]. In this report, significant tracer uptake was observed in lung tumors, but the tracer was rapidly cleared in normal patients. When the neutral Cu(II)-ATSM complex internalizes into cells, the metal ion is reduced to Cu(I). This reduction is more reversible when O_2_ is readily available, allowing the re-established Cu(II)-ATSM to leave the cell, but in hypoxic cells, the Cu(I)-ATSM remains trapped due to the reduction in lipophilicity. Furthermore, Cu(I) is less stable in the coordination environment established by the ATSM ligand, causing the release of the metal ion and re-chelation by chaperone proteins. For the past twenty years, [^60/61/62/64^Cu]Cu-ATSM tracers have been used to identify hypoxic tumors as well as monitor patients’ responses to radioimmunotherapy in multiple cancers [93]. However, the accumulation of [^60/61/62/64^Cu]Cu-ATSM is not always well correlated with hypoxic conditions. It has been reported that an increase in [^64^Cu]Cu-ATSM signal was observed by altering NAD(P)H metabolism in normoxic cells [94]. This issue has also been encountered clinically in the case of prostate cancer due to the tumors’ ability to restore redox balance [95].

### 2.5. Peptide-Based Radiopharmaceuticals for Targeted Cancer Imaging and Theranostics

Whereas radiolabeled nucleotide and amino acid mimics the function as PET tracers by becoming incorporated in metabolic processes elevated in cancer cells, entire peptide sequences and functionalized peptides can be used to target specific proteins. Such targets are commonly present on the cell membrane, providing facile access by the peptide-containing radiopharmaceutical. Such tracers can also be modified for radiolabeling by different radionuclides while maintaining the targeting peptide component. This section details three proteins which have been successfully targeted for PET imaging of cancerous lesions by tracers featuring peptide vectors.

The hypothalamic hormone somatostatin (SST) is a peptide sequence of 14 amino acids that leads to the initiation to the field of peptide-based radiopharmaceuticals for both diagnostic and therapeutic purposes. It was identified in 1973 and recognized to inhibit the secretion of growth hormones [96]. In humans, there are five different somatostatin receptors (SSTR-1 to SSTR-5). As SSTRs are overexpressed in numerous cancers, PET imaging with radiolabeled SST analogs has become a topic of great interest.

[^111^In]In-DTPA-octreotide ([^111^In]In-OctreoScan) [97] and [^99m^Tc]Tc-depreotide (NeoTect™) [98] are the first peptide-based radiopharmaceuticals approved for clinical use in the US and Europe. However, because of their short biological half-lives (1.7–1.9 h) and inability to bind all SSTRs, other analogues derived from octreotide analogues have been developed to improve tracer pharmacokinetics and targeting specificity [99,100]. These efforts were also coupled with the incorporation of positron-emitting radionuclides to acquire PET images with greater resolution compared to those obtained using SPECT. To date, ^64^Cu- and ^68^Ga-labeled DOTA-Tyr^3^-octreotide (DOTATOC) and DOTA-Tyr^3^-octreotate (DOTATATE) have gained significant clinical relevance for PET imaging and peptide receptor radiotherapy in patients with neuroendocrine tumors (NETs) [101]. Comparative clinical studies of [^68^Ga]Ga-DOTATATE with [^18^F]FDG [102] and [^111^In]In-DTPA-octreotide [103,104] have supported [^68^Ga]Ga-DOTATATE’s superiority in identifying lesions. Substitution of ^68^Ga for ^64^Cu in a clinical setting was first described in 2012 [105], with later studies supporting its prognostic value in predicting progression-free survival (though not overall survival) [106]. Even though the affinity of [^68^Ga]Ga-DOTATATE for SSTR-2 was found to be approximately 10 times greater than that of [^68^Ga]Ga-DOTATOC [107], they performed similarly in detecting NETs in 40 patients and [^68^Ga]Ga-DOTATOC displayed greater signal intensity [108]. More recently, changing the ^68^Ga chelator from DOTA to DATA to form [^68^Ga]Ga-DATATOC allowed for a more convenient synthesis in a radiopharmacy setting while performing comparably to [^68^Ga]Ga-DOTATOC in clinical PET/CT experiments [109]. In addition, ^18^F-labeled octreotide analogues have recently been developed for clinical applications due to the higher production yields and greater potential patient population that can be served with ^18^F compared with ^68^Ga [110,111]. Recent milestones in the clinical success of [^64^Cu/^68^Ga]Cu/Ga-DOTATATE and [^68^Ga]Ga-DOTATOC were reached with their FDA approval for routine clinical use in the localization of somatostatin receptor positive neuroendocrine tumors in 2020, 2016, and 2019, respectively.

Following the success of SSTR-targeted radiopharmaceuticals, prostate-specific membrane antigen (PSMA)-based PET imaging is emerging as an important tool for the management of prostate cancer patients. Among all PSMA-targeted radiopharmaceuticals, [^68^Ga]Ga-PSMA-11 (Figure 8) is perhaps the most investigated PET agent for imaging prostate cancer. Afshar-Oromieh et al. initiated the evaluation of [^68^Ga]Ga-PSMA-11 as a novel PET agent for prostate cancer [112]. The researchers found that [^68^Ga]Ga-PSMA-11 detects recurrent and metastatic prostate cancer through binding to the extracellular domain of PSMA followed by internalization of the compound/agent into the cell. Since then, multiple studies have confirmed the superiority of [^68^Ga]Ga-PSMA-11 to other conventional PET tracers, such as [^11^C]choline, [^18^F]fluorocholine, and [^18^F]FACBC, for imaging patients with prostate cancer [113]. In 2020, [^68^Ga]Ga-PSMA-11 was approved by the FDA to image recurrent metastatic prostate cancer [114]. Continuing retrospective studies have reported using this successful radiopharmaceutical in imaging and influencing the management of other cancers such as primary clear-cell renal cell and hepatocellular carcinoma as well [115,116,117].

The success of [^68^Ga]Ga-PSMA-11 further inspired the continued development of PSMA-targeting PET tracers for imaging prostate cancer. [^18^F]PSMA-1007 (Figure 8) and 2-(3-{1-carboxy-5-[(6-[^18^F]fluoro-pyridine-3-carbonyl)-amino]-pentyl}acc-ureido)-pentanedioic acid ([^18^F]DCFPyL) (Figure 8) have received the most clinical attention, owing to the convenient availability of ^18^F and the large-scale production of these two tracers. Of note, [^18^F]DCFPyL has recently been approved by the FDA [118]. According to the meta-analysis by Treglia et al. [119], which included six investigations with a total of 645 patients with biochemically recurrent prostate cancer, [^18^F]PSMA-1007 and [^18^F]DCFPyL have similar detection rates dependent on PSA levels, and each has noted advantages over [^68^Ga]Ga-PSMA-11. For example, as [^18^F]PSMA-1007 has demonstrated characteristics indicative of hepatobiliary excretion, imaging with this tracer results in lower background signal near the prostate bed compared to imaging with [^68^Ga]Ga-PSMA-11, which is excreted through a urinary pathway. Both [^18^F]PSMA-1007 and [^18^F]DCFPyL have exhibited excellent image quality compared to [^68^G]Ga-PSMA-11, including in the cases of local recurrence (Figure 9).

In addition to SSTR and PSMA, fibroblast activation protein (FAP) appears to be another promising biomarker that has recently received great attention. FAP is involved in numerous pathological processes with the complete number and identity still a topic of investigation, though its role in wound healing and inflammation is reported [121,122,123]. FAP can also serve as an important target for oncological imaging and therapy due to its near-universal overexpression in a variety of cancers, contributing up to 90% of the gross tumor mass [124]. PET tracers have since been developed to target this commonly encountered protein with advancements originating from FAP-specific inhibitor (FAPI) anticancer drugs. These inhibitors were initially designed from a quinoline-containing scaffold with an additional cyanopyrrolidine moiety [125].

A series of DOTA-containing FAPI congeners were synthesized and compared according to their specificity for FAP and internalizing properties, resulting in the designation of FAPI-04 as the most promising for PET imaging [126,127]. Significant uptake was observed when using [^68^Ga]Ga-FAPI-04 to detect multiple cancers including breast, esophagus, lung, pancreatic, head–neck, and colorectal cancers—the tumor-to-background contrast ratio being enhanced three-fold in the intermediate uptake group (Figure 10) [128]. In addition, the high and selective tumor uptake of [^68^Ga]Ga-FAPI-46 has recently been confirmed in patients with advanced lung cancer [129]. These encouraging results are likely to prompt additional studies utilizing [^68^Ga]Ga-FAPI PET/CT for tumor characterization and progress monitoring during radiopharmaceutical therapy. ^18^F-labeled counterparts (FAPI-42 and FAPI-74) are currently being developed for centralized production to mitigate the potential challenges of commercial distribution [130,131].

The peptide sequences and mimics present in PET tracers imaging the overexpression of SST, PSMA, and FAP function as targeting components for these specific proteins; therefore, the overall function of the radiopharmaceutical can be changed (such as from imaging to therapy) by changing the radionuclide. Targeted radionuclide therapy is an evolving and promising approach, in which therapeutic radionuclides that can emit particles with high linear energy transfer (LET) (e.g., α, β^−^, and Auger electrons) are carried by a targeting molecule to deliver radiation to cancer cells while sparing normal tissue. Along with targeted radionuclide therapy, the concept of theranostics has also emerged to incorporate imaging diagnostics and radionuclide therapy using chemically identical radiopharmaceuticals with nuclides of the same element [132,133]. Highly localized radiotherapy using systemically administered radioligands has gradually become a standard treatment option for many solid tumors that cannot be successfully excised surgically.

## 3. Conclusions

PET radiopharmaceuticals have experienced tremendous growth during the last decade as a result of considerable progress made in PET technologies and image analysis methodologies. With the foundation of PET imaging in oncology established by the glucose derivative [^18^F]FDG, a variety of biological targets presented in this article have also received great attention by researchers to direct signals towards tumor sites. The future of radiopharmaceuticals is bright. In the last few years, we have seen a rapid increase in FDA approvals or clinical trials for radiopharmaceuticals. With substantial growth in the use of radiopharmaceuticals, the excitement is beyond individual diagnostic imaging, but is rather for adapting theranostic concepts that could further improve the clinical experience for patients.

## Figures and Tables

**Figure 1 molecules-27-06790-f001:**
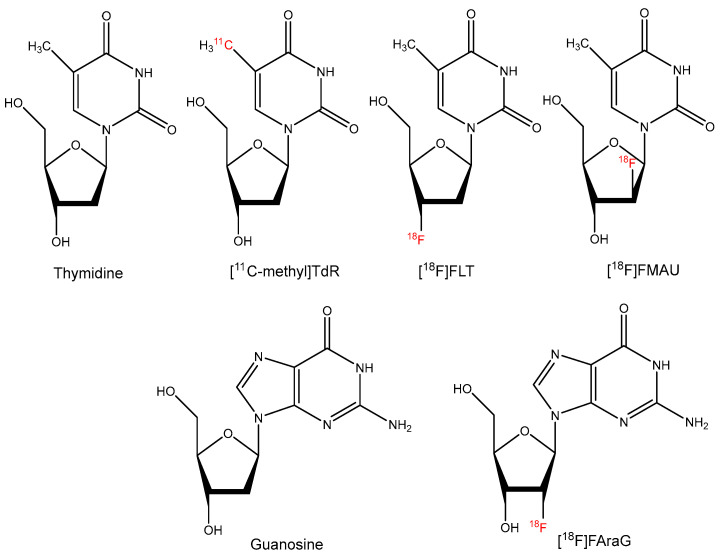
Chemical structures of nucleotide-based PET radiopharmaceuticals.

**Figure 2 molecules-27-06790-f002:**
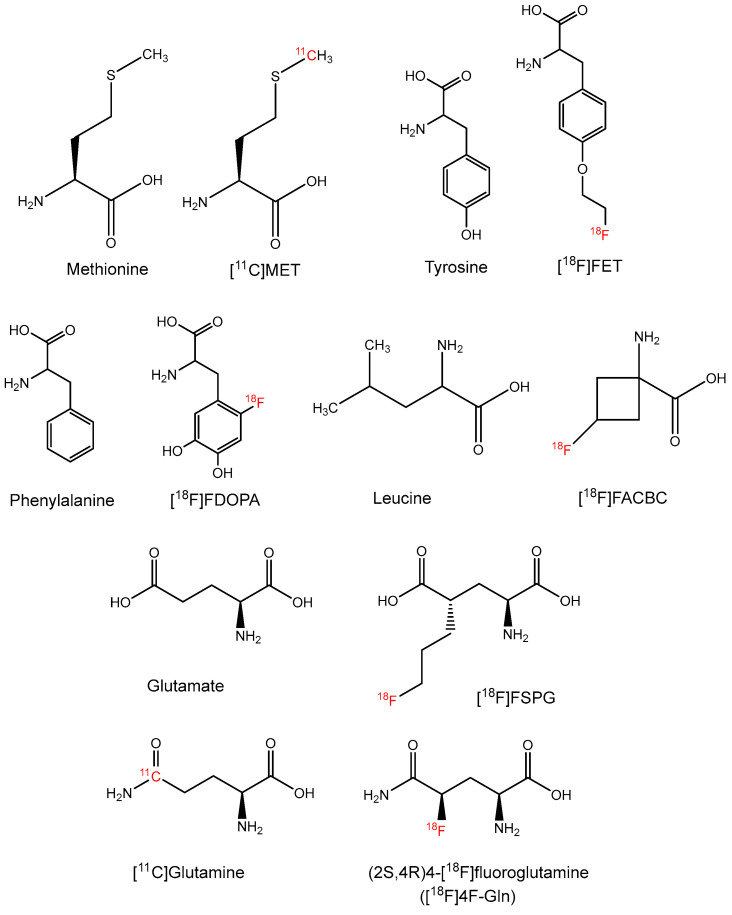
Chemical structures of amino acid-based PET radiopharmaceuticals.

**Figure 3 molecules-27-06790-f003:**
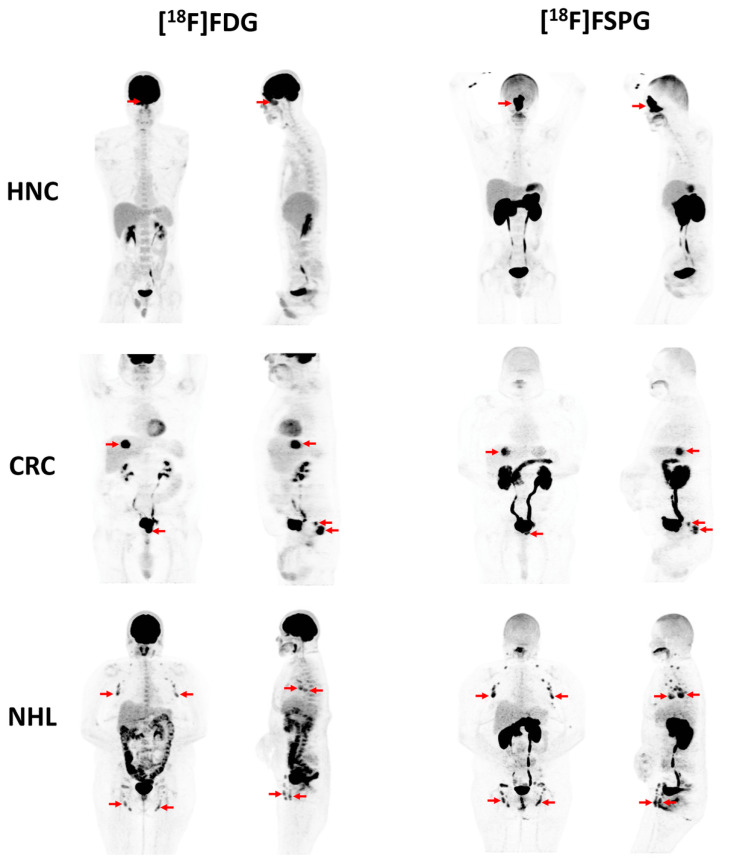
Whole-body maximum intensity projection (MIP) images are shown of representative participants with head and neck cancer (HNC, top), colorectal cancer (CRC, middle), or non-Hodgkin lymphoma (NHL, bottom). The [^18^F]FDG PET images (front and side views) are shown on the left, and the [^18^F]FSPG PET images (front and side views) are shown on the right. Red arrows indicate sites of malignancy on each scan. Reprinted with permission from EJNMMI Research, originally published by Park et al. [38].

**Figure 4 molecules-27-06790-f004:**
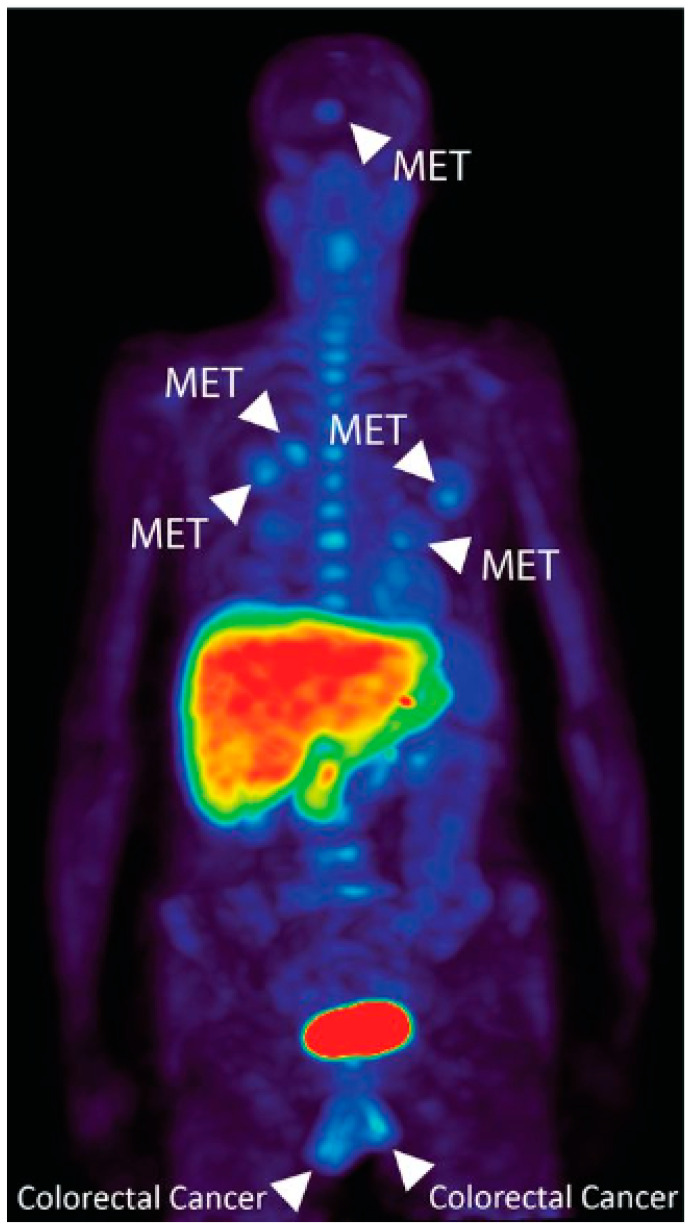
Representative [^11^C]glutamine PET imaging in patient with metastatic colorectal cancer. Reprinted with permission from the Journal of Nuclear Medicine, originally published by Cohen et al. [40].

**Figure 5 molecules-27-06790-f005:**
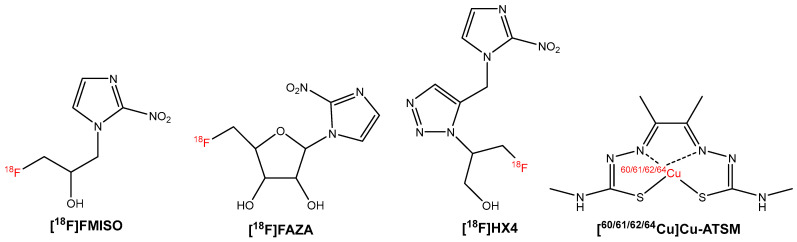
Chemical structures of [^18^F]FMISO, [^18^F]FAZA, [^18^F]HX4, and [^60/61/62/64^Cu]Cu-ATSM.

**Figure 6 molecules-27-06790-f006:**
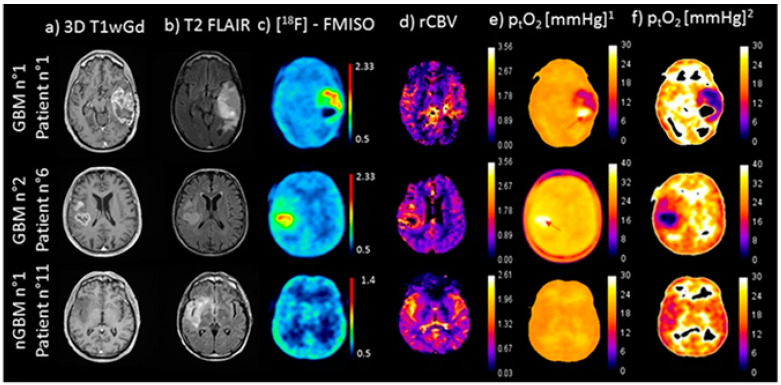
Two patients with glioblastoma (GBM, top, and middle) and one patient with less aggressive glioma (nGBM, bottom) were imaged with multiparametric magnetic resonance imaging (MRI) (anatomical and perfusion weighted imaging (PWI)) and [^18^F]FMISO PET. Representative axial three-dimensional T1-weighted gadolinium enhanced (3D T1wGd) MRI (**a**), fluid-attenuated inversion recovery (FLAIR) MRI (**b**), [^18^F]FMISO PET (**c**), and PWI-derived relative cerebral blood volume (rCBV) maps (**d**) are shown. Partial oxygen pressure in tissue (p_t_O_2_) is depicted in representative absolute maps (**e**) and adjusted maps with non-linear regression (**f**). The tumor-to-background ratio for [^18^F]FMISO signal was 1.4 ± 0.2 for GBM and 0.9 ± 0.1 for nGBM (*p* < 0.001). The uptake of [^18^F]FMISO correlates with GBM tumor site and volumes with diminished p_t_O_2_; reprinted with permission from Scientific Reports, originally published by Chakhoyan et al. [61].

**Figure 7 molecules-27-06790-f007:**
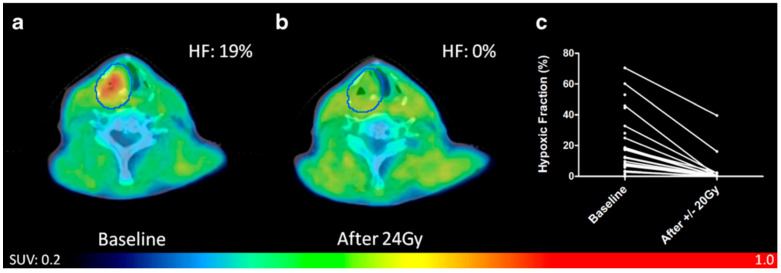
A patient with a T3N2bMx squamous cell carcinoma of the hypopharynx was imaged by [^18^F]HX4 PET/CT (**a**) before and (**b**) after chemoradiotherapy and displayed a reduction in hypoxic fractions in primary tumors and lymph nodes, as depicted in (**c**). Reprinted with permission from the European Journal of Nuclear Medicine and Molecular Imaging, originally published by Zegers et al. [85].

**Figure 8 molecules-27-06790-f008:**
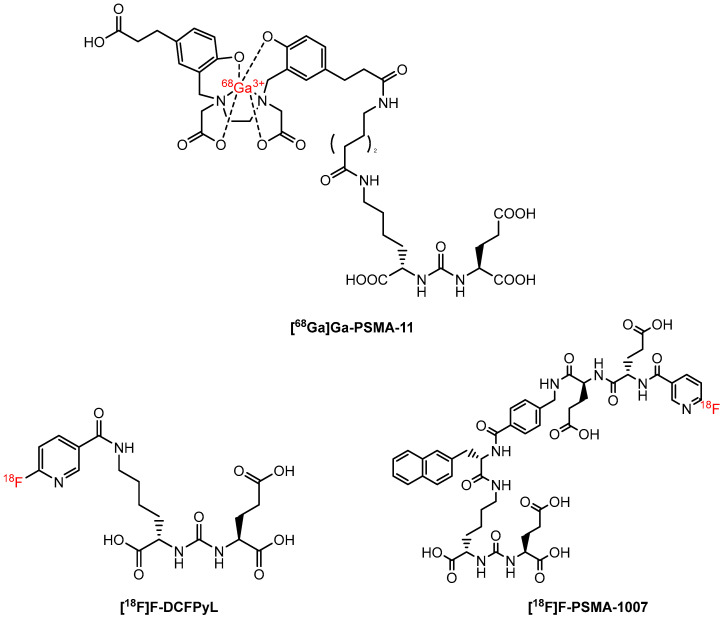
Chemical structures of [^68^Ga]Ga-PSMA-11, [^18^F]DCFPyL, and [^18^F]PSMA-1007.

**Figure 9 molecules-27-06790-f009:**
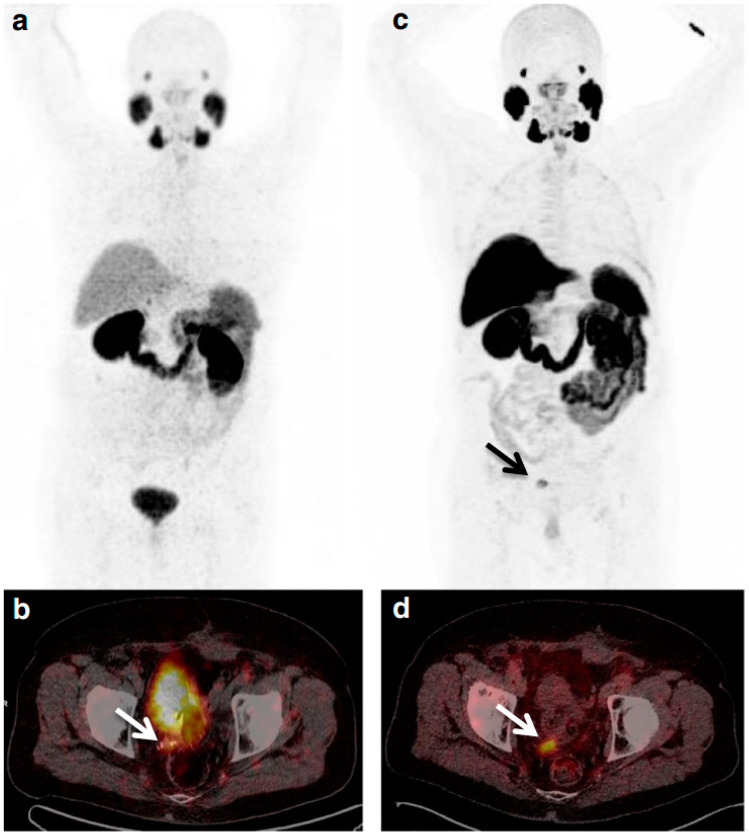
Images of a 74-year-old prostate cancer patient after radical prostatectomy (Gleason score 9) with biochemical recurrence (PSA: 2.1 ng/dL). Images (**a**,**b**) show [^68^Ga]Ga-PSMA-11 PET-CT ((**a**): maximum-intensity projection, MIP; (**b**): fused axial PET-CT image). Arrows show minimal pararectal uptake close to the bladder and the ureter, for which clinical decision-making is problematic. Images (**c**,**d**) show [^18^F]PSMA-1007 PET-CT of the same patient ((**c**): MIP, (**d**): fused axial PET-CT image). Arrows show unequivocal focal uptake representing a local recurrence, with high contrast (maximum standard uptake value: 9.9), with no distracting ureteral or vesical excretion activity. [^18^F]PSMA-1007 seems to be superior to [^68^Ga]Ga-PSMA-11 in cases of biochemical recurrence and unclear lesions close to the ureter or urinary bladder. Reprinted with permission from the European Journal of Nuclear Medicine and Molecular Imaging, originally published by Rahbar et al. [120].

**Figure 10 molecules-27-06790-f010:**
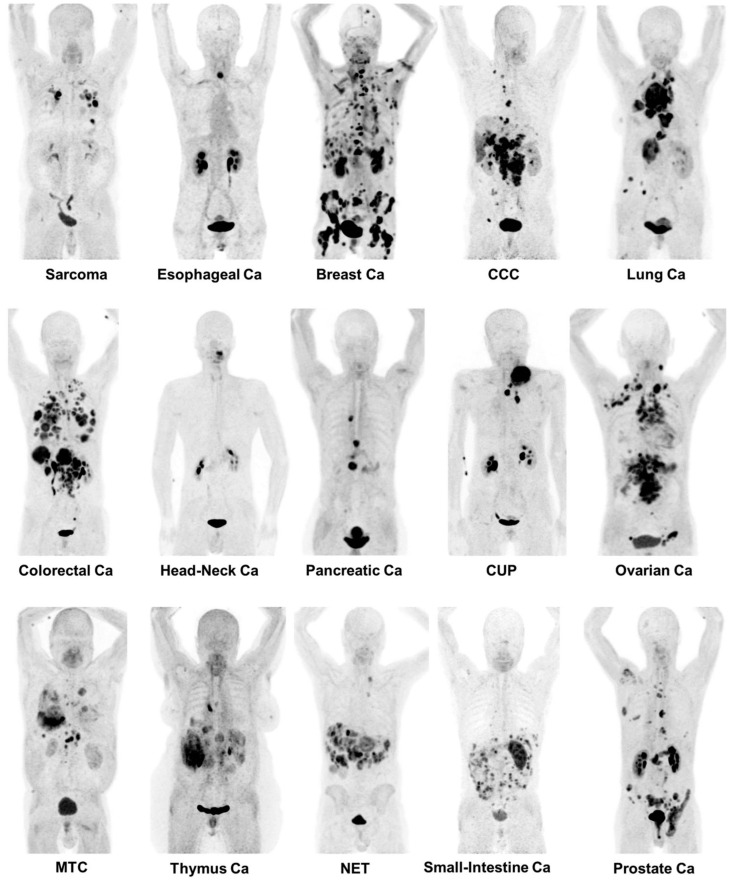
Maximum-intensity projections of [^68^Ga]Ga-FAPI PET/CT in patients reflecting 15 different histologically proven tumor entities (sorted by uptake in descending order). Ca = cancer; CCC = cholangiocellular carcinoma; CUP = carcinoma of unknown primary; MTC = medullary thyroid cancer; NET = neuroendocrine tumor. Reprinted with permission from the Journal of Nuclear Medicine, originally published by Kratochwil et al. [128].

## Data Availability

Not applicable.

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
