# Peer review of "PET Oncological Radiopharmaceuticals: Current Status and Perspectives"

_molecules, 2022, doi:10.3390/molecules27206790_

Round 1

Reviewer 1 Report

Review on the manuscript entitled “PET Oncological Radiopharmaceuticals: Current Status and Perspectives” written by Mai Lin et al.

The manuscript is based on PET imaging and its development over the years with focus on the PET radiopharmaceutical and the application particularly in the oncology domain.

The manuscript is well written, I would recommend checking for fine spelling and answering the queries presented below.

Please add the criteria used for selecting the articles utilized in the manuscript.

Introduction – Provides significant content related to the proposed topic.

Content – Is organized in sections and subsection, has fluency in ideas, should be improved with articles published in the last 5-10 years.

Line 138 – Please remove the duplicate text.

In the chemical structures, I recommend underling the positron emitter (maybe using a color) especially to differentiate from the derived structure.

Figure 3 and 6 should be replaced with higher resolution images.

Conclusions – Are correlated with the presented information.

References – Some references date back to the 1950’s, if possible I recommend replacing them with works that are more recent.

 I consider that more than 75% of the references should be from the last 5-10 years, according to the title of the manuscript that implies the “current status and perspectives” for the approached area.

Reviewer 2 Report

The authors present a review on PET oncological radiopharmaceuticals. The review is well written, clearly presented, with a lot of different studies, and allows to have a clear vision of the state of the art of PET imaging in oncology. I thus recommend the publication of the review without any modifications.

Round 2

Reviewer 1 Report

The authors improved the manuscript by reorganizing the text and underling the application of the PET imaging.

All the queries were replied but I recommend the authors to carefully check the manuscript for fine mistakes and misspell.

Author Response

We appreciate the kind words from the reviewer. We have checked the manuscript and corrected mistakes.misspell accordingly.